# An Updated Review of Management of Resectable Stage III NSCLC in the Era of Neoadjuvant Immunotherapy

**DOI:** 10.3390/cancers16071302

**Published:** 2024-03-27

**Authors:** Saurav Verma, Daniel Breadner, Abhenil Mittal, David A. Palma, Rahul Nayak, Jacques Raphael, Mark Vincent

**Affiliations:** 1Division of Medical Oncology, Department of Oncology, Schulich School of Medicine & Dentistry, Western University, London, ON N6A 3K7, Canada; saurav.verma@lhsc.on.ca (S.V.); daniel.breadner@lhsc.on.ca (D.B.); jacques.raphael@lhsc.on.ca (J.R.); 2London Regional Cancer Program, London Health Sciences Centre, London, ON N6A 5W9, Canada; david.palma@lhsc.on.ca (D.A.P.); rahul.nayak@lhsc.on.ca (R.N.); 3Division of Medical Oncology, Northeast Cancer Centre, Ramsey Lake Health Centre, Sudbury, ON P3E 5J1, Canada; drabhenil@gmail.com; 4Division of Radiation Oncology, Department of Oncology, Schulich School of Medicine & Dentistry, Western University, London, ON N6A 3K7, Canada; 5Division of Thoracic Surgery, Department of Oncology, Schulich School of Medicine & Dentistry, Western University, London, ON N6A 3K7, Canada

**Keywords:** stage III, resectable, locally advanced, NSCLC, lung cancer, immunotherapy, perioperative, neoadjuvant, immune checkpoint inhibitors, ICI

## Abstract

**Simple Summary:**

The recent evidence shows that patients with resectable non-small cell lung cancer (NSCLC) benefit from the addition of immunotherapy to chemotherapy before surgery. This approach improves pathological response rates and overall survival in these patients. However, there are many unanswered questions, especially in the context of heterogenous stage III NSCLC. In this review we discuss the new evidence, evolving definitions of resectability, controversies, possible salvage treatments and the ongoing research in this rapidly changing space. We highlight that multimodal treatment for stage III NSCLC should be individualized based on patient and tumor variables. Neoadjuvant immunotherapy and chemotherapy should be discussed with all patients with resectable (>4 cm and/or node positive, excluding N3) NSCLC.

**Abstract:**

Immune-checkpoint inhibitors (ICIs) have an established role in the treatment of locally advanced and metastatic non-small cell lung cancer (NSCLC). ICIs have now entered the paradigm of early-stage NSCLC. The recent evidence shows that the addition of ICI to neoadjuvant chemotherapy improves the pathological complete response (pCR) rate and survival rate in early-stage resectable NSCLC and is now a standard of care option in this setting. In this regard, stage III NSCLC merits special consideration, as it is heterogenous and requires a multidisciplinary approach to management. As the neoadjuvant approach is being adopted widely, new challenges have emerged and the boundaries for resectability are being re-examined. Consequently, it is ever more important to carefully individualize the treatment strategy for each patient with resectable stage III NSCLC. In this review, we discuss the recent literature in this field with particular focus on evolving definitions of resectability, T4 disease, N2 disease (single and multi-station), and nodal downstaging. We also highlight the controversy around adjuvant treatment in this setting and discuss the selection of patients for adjuvant treatment, options of salvage, and next line treatment in cases of progression on/after neoadjuvant treatment or after R2 resection. We will conclude with a brief discussion of predictive biomarkers, predictive models, ongoing studies, and directions for future research in this space.

## 1. Introduction

Lung cancer is the leading cause of global cancer-related mortality [1]. Non-small cell lung cancer (NSCLC) contributes to 80–85% of all lung cancer cases. Of these, approximately 20–25% are diagnosed with stage III disease [2,3]. Despite recent advances the outcomes remain suboptimal, with a 5-year survival rate of 10–42% for stage III NSCLC, depending on the sub-stage [2,4,5]. Patients with stage III NSCLC represent a heterogenous population and the treatment plan is dependent on multiple factors, including the patient’s ability to achieve an R0 resection, ability to tolerate resection, and ability to tolerate systemic therapy [6]. The conventional treatment strategy for a patient with resectable stage III NSCLC consists of surgery paired with adjuvant or neoadjuvant (NA) systemic therapy, with utilization of radiation in selected cases. The 5-year absolute benefit in overall survival (OS) with adjuvant/NA chemotherapy is modest, at 5% [7,8]. Recent trials have shown that the addition of immune checkpoint inhibitors (ICIs) to chemotherapy in the NA setting improves response and survival outcomes in intention to treat (ITT) populations [9,10]. Recent data have been evaluated in a meta-analysis, showing that NA/perioperative immuno-chemotherapy (ICI-chemo) results in significant improvement in pathological complete response (pCR) rate (risk ratio (RR): 5.66, 95% CI 3.48–9.18), event-free survival (EFS), OS, major pathological response (MPR) rate, and R0 resection rate compared to chemotherapy alone [11,12]. NA nivolumab with chemotherapy is now approved by the Food and Drug Administration (FDA) and European Medicines Agency (EMA) in resectable, stage IB-IIIA (American Joint Committee on Cancer (AJCC), 7th Edition) NSCLC, based on the CheckMate 816 trial. This trial demonstrated improved pCR rate and EFS (OS data are immature) [10]. Furthermore, perioperative pembrolizumab (NA pembrolizumab-chemo followed by surgery and adjuvant pembrolizumab) has shown an improved OS compared to NA chemotherapy alone in patients with resectable NSCLC in the Keynote 671 trial and was recently approved by the FDA [13]. NA ICI-chemo is now being adopted widely and has been included as a standard of care option in the NCCN, ESMO and ASCO guidelines for resectable NSCLC [14,15]. Other studies have replicated the findings of benefit with different ICIs, with some studies adopting a perioperative approach [9,13,16]. Notably, these studies had differences in design (either NA or perioperative based, number of cycles) and patient population.

Although the new approach of NA ICI-chemo in resectable NSCLC is promising, many issues remain unsettled. The parameters of resectability are evolving and require a reassessment of the definition of resectability in the current era of immunotherapy. More data are needed for patients with T4 and N2 disease (especially multi-station N2) to inform the optimal treatment strategy. It is still unknown if nodal downstaging can make unresectable disease resectable, and if resected, would the outcomes be comparable to concurrent chemoradiation (CTRT) followed by consolidation ICI based on the PACIFIC trial [5]. This narrative review aims at dissecting the recent literature in this setting with the aims of highlighting the challenges, discussing possible solutions, and outlining the directions of ongoing and future research. We searched PubMed for articles in the English language from 2018 to 2023 using the keywords neoadjuvant immunotherapy, early-stage NSCLC and resectable NSCLC. We also searched clinicaltrials.gov with the keywords neoadjuvant immunotherapy and NSCLC.

## 2. Resectability in NSCLC in the Era of Neoadjuvant Immunotherapy

Stage III NSCLC is a heterogenous condition, with various combinations of tumor size and local nodal involvement [17]. Recently, the final proposal for the 9th Edition TNM classification was presented, which proposed the splitting of N2 into single-station N2 (N2a) and multiple station N2 (N2b). This is a clinically relevant categorization for treatment decisions and prognosis in these patients [18]. The staging as per the 8th and proposed 9th edition TNM is presented in Table 1 [17,18]. The optimal treatment strategy requires multimodal treatment, and it is recommended that these cases be discussed in a multidisciplinary tumor board (MDT) to reach a consensus on the treatment plan. Stage IIIA has been an area of controversy for surgical resection, with many investigators regarding single-station N2 disease (especially in the upper mediastinum) as acceptable for surgery, but with anything beyond that being generally more favored for CTRT and consolidation durvalumab (e.g., multi-station N2, extra-nodal extension, lower mediastinal single station, or T4 disease invading vital structures, etc.) [19,20]. Stage IIIB is generally regarded as unsuitable for resection; with occasional exceptions as adjudicated by MDTs [21]. In the real-world setting, patient selection is more complex and is influenced by institutional practices and preferences. The definition of resectability is based on the assessment of whether an R0 resection can be achieved, as determined by an MDT [21,22]. Complete resection (R0) is achieved when resection margins are negative for malignancy and systematic nodal sampling has been performed. Additionally, the highest lymph node station should be negative and there should not be any extracapsular nodal involvement [22]. ASCO guidelines recommended the following conditions to be met for patients with NSCLC for induction therapy followed by surgery: (1) a complete resection (R0) of the primary tumor and involved lymph nodes is deemed possible; (2) N3 lymph nodes are deemed to be not involved by multidisciplinary consensus; and (3) perioperative (90-day) mortality is expected to be low (≤5%) [21]. In addition to resectability, careful consideration of variables affecting fitness for surgery, like age, pulmonary function, cardiovascular fitness, nutrition, and performance status are paramount [23].

## 3. Evidence for Neoadjuvant ICI-Chemotherapy in Stage III NSCLC

There has been a rapid accumulation of evidence for NA ICI-chemo (±adjuvant ICI after surgery) in resectable NSCLC (Figure 1, Table 2 and Table 3). The NADIM II and CheckMate 816 trials have shown improvement in pCR and progression free survival (PFS)/EFS with NA ICI-chemo [9,10]. Recently, Keynote 671 has shown an improvement in EFS along with an OS benefit [13,24]. CheckMate 77T and AEGEAN have shown similar findings, although OS data are immature [25,26].

### 3.1. CheckMate 816

CheckMate 816 was an open-label, phase 3 trial, which included patients with stage IB to IIIA (AJCC 7th edition) resectable NSCLC [10]. The patients were randomly assigned to receive nivolumab plus platinum-doublet chemotherapy (experimental arm) or chemotherapy alone (control arm), followed by surgery. In the experimental arm patients could receive carboplatin-paclitaxel (any histology), gemcitabine-cisplatin (squamous histology), or pemetrexed-cisplatin (non-squamous histology). In the control arm patients could receive docetaxel or vinorelbine with cisplatin, in addition to chemotherapy regimens allowed in experimental arm. Carboplatin was allowed in patients ineligible for cisplatin. The median follow-up period was 41.4 months [27,28]. The recent 3-year update on the trial reported an improvement in the median EFS, not reached (NR) (95% confidence interval (CI), 31.6 to NR) in the experimental arm, and 21.1 months (95% CI, 14.8 to 42.1) in the control arm (HR, 0.68; 95% CI, 0.49 to 0.93). The pCR rates were 24.0% (95% CI, 18.0 to 31.0) and 2.2% (95% CI, 0.6 to 5.6), respectively (odds ratio (OR), 13.94; 99% CI, 3.49 to 55.75; *p* < 0.001). The EFS was improved regardless of surgical approach, and the HR for EFS was 0.61 (95% CI, 0.28–1.29) and 0.74 (95% CI, 0.48–1.13) for minimally invasive surgery and thoracotomy/conversion, respectively. The HR for EFS for lobectomy was 0.62 (95% CI, 0.40 to 0.96). Interestingly, locoregional recurrence was similar, at 22% in the experimental arm (n = 29) vs. 19% in the control arm (n = 28); however, there was a trend towards decreased distant recurrence in the experimental arm, at 10% (n = 15) vs. 22% (n = 30) and decreased intracranial recurrence (4% (n = 6) vs. 13% (n = 17)). The median OS was NR, HR 0.62 (99.34% CI 0.36–1.05; *p* = 0.0124) with a trend towards benefit and separation of Kaplan-Meier curves, but the OS data are immature. Surgery-related adverse events (AEs) were similar in both arms. Grade 5 surgery-related AEs (one each due to pulmonary embolism and aortic rupture in the experimental arm) were deemed to be unrelated to ICIs. Interestingly, the proportion of patients who had surgery was higher in the experimental arm (83.2 vs. 75.4%) with a lesser median duration of surgery (184 vs. 217 min). Among the patients who could not undergo surgery in the experimental arm, only two had CTRT and adjuvant ICI, while the rest either had radiotherapy alone or systemic therapy alone (chemotherapy or ICI-chemo or ICI alone).

63.7% of patients had stage III disease at baseline. The disease stage at entry was IIIA in 63.1% (n = 113) and 64.2% (n = 115), and following NA treatment the disease stage was IIIA in 45.3% and 48.6% of patients in the experimental and control arms, respectively. In the absence of individual patient data, it is hard to determine whether the disproportionate decrease in stage IIIA patients after systemic treatment in the two arms is predominantly due to more downstaging in the experimental arm or relatively more upstaging/progression in the control arm. There were 23% and 40.7% of patients with stage IIIA disease who had a pCR and MPR in the experimental arm vs. 0.9% and 9.6% in the control arm. There were 21.8% and 39.1% of patients with stage IIIA disease with lymph node involvement who had a pCR and MPR in the experimental arm vs. 0% and 9.7% in the control arm, which points to a possibility that the pathological response may be independent of the presence or absence of nodal disease burden. The median residual tumor volume (depth of pathological response as measured by percentage of residual cells in the primary tumor bed) was 8% (interquartile range (IQR), 0–80) in the experimental arm and 70% (IQR, 40–90) in the control arm, for patients with stage IIIA disease.

Based on evidence from this trial, in May 2022, the FDA approved the combination of ICI-chemo used in CheckMate 816 for stage IB-IIIA NSCLC. The combination has been adopted widely, albeit with some variations in the degree of its use in the eligible population.

### 3.2. NADIM II

NADIM II was an open-label, phase 2 RCT, in patients with resectable stage IIIA or IIIB NSCLC. Patients (n = 86) received either NA nivolumab plus paclitaxel-carboplatin chemotherapy or paclitaxel-carboplatin alone, followed by surgery. In the trial, tumor resectability was reassessed after NA treatment by an MDT board based on the possibility of obtaining R0 resection. Patients in the experimental group who had an R0 resection received adjuvant treatment with nivolumab for six months. Patients with N3 nodal stage cancer, and whose tumors were positive for EGFR/ALK mutations or rearrangements, were excluded. Two thirds of the patients had pathologically proven N2 disease, including involvement of multiple N2 stations.

The primary endpoint (pCR) occurred in 37% and 7% of patients (relative risk (RR), 5.34; 95% CI, 1.34 to 21.23; *p* = 0.02) and surgery was performed in 93% and 69% of patients (RR, 1.35; 95% CI, 1.05 to 1.74) in the experimental and control groups, respectively. At 24 months, the PFS was 67.2% vs. 40.9% (HR, 0.47; 95% CI, 0.25 to 0.88) and the OS was 85.0% vs. 63.6% (HR, 0.43; 95% CI, 0.19 to 0.98) in the experimental and control arms, respectively. Grade 3 or 4 AEs occurred in 11 patients in the experimental group (19%) and three patients in the control group (10%). Patients had surgery 7 to 9 weeks after the first day of the third cycle. Distant recurrence occurred as the first event in 18% and 28% patients in the experimental group and control group, while locoregional recurrence occurred in 12% and 31%, respectively. CNS relapses occurred in 5% and 14% patients in the experimental group and control group, respectively. All the patients who had a pCR were free from recurrence and were alive at the time of data cut-off in the last report. Again, nodal involvement was not associated with differences in outcomes of response or survival.

Notably, 7% (n = 4) of patients in the experimental group and 31% (n = 9) in the control group did not undergo surgery. One patient who underwent right pneumonectomy in the experimental group died due to surgical complications 13 days after surgery, although no additional deaths were noted at either 30 days or 90 days from surgery. It was noted that among the patients who had R0 resections, those who completed adjuvant treatment had better survival outcomes than those who did not complete adjuvant treatment (HR 0.38 (0.14–1.06) for PFS; HR 0.29 (0.05–1.76) for OS), even after excluding all patients who had had a pCR from this analysis. Treatment-related AEs (TRAEs) of any grade leading to discontinuation of NA treatment occurred in four and one patient in the experimental group and control group, respectively. There were no delays in surgery due to any AEs from systemic treatment. In addition, worse survival outcomes were not observed among patients with N2 disease than among those with N0 or N1 disease in the experimental arm.

Neither tumor mutation burden (TMB) nor programmed cell death ligand-1 (PD-L1) status were predictive of survival. However, low pretreatment levels of ctDNA were significantly associated with improved PFS and OS (HR, 0.20; 95% CI, 0.06 to 0.63, and HR, 0.07; 95% CI, 0.01 to 0.39, respectively). Also, 67% patients in the experimental group and 44% in the control group were ctDNA-negative after NA treatment. Clinical responses according to RECIST v1.1 criteria did not predict survival outcomes. However, undetectable ctDNA levels after NA treatment were significantly associated with better PFS and OS (HR, 0.26; 95% CI, 0.07 to 0.93, and HR, 0.04; 95% CI, 0.00 to 0.55, respectively). The Harrell’s concordance index (C-index) was superior for predicting OS for ctDNA levels after NA treatment (0.82) to the RECIST criteria (0.72) [29].

### 3.3. TD-FOREKNOW

In this randomized phase 2 clinical trial, patients with resectable stage IIIA or IIIB (T3N2) NSCLC (EGFR and ALK mutation/alteration included) were treated with three cycles of camrelizumab (an anti-PD1 ICI) plus chemotherapy or chemotherapy alone followed by surgery. The primary endpoint (pCR rate) was 32.6% with camrelizumab plus chemotherapy vs. 8.9% with chemotherapy alone. The median EFS and disease-free survival (DFS) rates were NR. 79.1% and 68.9% of patients with N2 disease, and 13.9% vs. 22.2% of those who had T4 disease at baseline, in the experimental and control arms, respectively. Despite a higher number of patients with N2 disease, pCR was impressive, at 32.6% in the experimental arm. Information on multi-station/bulky or single-station N2 lymph nodes was not available. The trial used nab-paclitaxel with platinum (cisplatin/carboplatin/nedaplatin) to avoid the need for corticosteroids. During sample size calculation, the pCR rates were assumed to be 50% in the experimental arm and 18% in the control arm, which were overestimates with respect to the results achieved. Notably, the pCR benefit was consistent across subgroups, irrespective of PD-L1 tumor proportion score (TPS), status (<1 or ≥1), or stage (IIIA vs. IIIB). In total, 52% of patients had an unknown PD-L1 status. Inclusion criteria were revised from including stage II-IIIB to IIIA-IIIB. The occurrence of more TRAEs in the experimental group was explained by a high incidence of camrelizumab-related reactive cutaneous capillary endothelial proliferation (RCCEP, a common immune-related AE (irAE) related to camrelizumab). All irAEs were either grade 1 or 2. There were no grade 3 irAEs, although irAEs of any grade occurred in 53.5% patients. Any grade surgery-related AEs occurred in 40% vs. 33.3%, with one death due to thoracic trauma (deemed unrelated to the experimental treatment) after one cycle of NA treatment [30].

### 3.4. AEGEAN

This was a double-blind phase 3 RCT assessing durvalumab in a perioperative setting. Patients (n = 802) with resectable NSCLC (stage II to IIIB (N2 nodes), AJCC 8th edition) were randomly assigned to either receive platinum-doublet chemotherapy plus durvalumab or chemotherapy plus placebo for four cycles before surgery, followed by adjuvant durvalumab or placebo for 12 cycles. Mediastinal lymph-node staging was performed at the discretion of the investigator. The protocol was amended to exclude patients with tumors classified as T4 for any reason other than size (>7 cm), whose planned surgery at enrollment was pneumonectomy, or who had documented test results that confirmed the presence of an EGFR mutation. The recent report presents an initial analysis of a modified ITT population (n = 740), excluding those with EGFR/ALK alterations. Notably, 71.3% and 70.3% patients in the experimental and control groups had stage III disease, with 49.5% having N2 disease (single or multi-station) in each arm. Approximately, ¼ of patients in each arm had stage IIIB disease (24% vs. 26.2%). The median duration of follow-up was 11.7 months. Interestingly, in this trial the numbers of patients who underwent surgery were similar in both arms (80.6 vs. 80.7%). In total, 6.4% of the patients received postoperative radiotherapy. The duration of EFS was improved in the durvalumab arm (HR 0.68, 95% CI, 0.53 to 0.88; *p* = 0.004), and EFS at 12-months was 73.4% vs. 64.5% (95% CI, 58.8 to 69.6). The pCR rates were 17.2% vs. 4.3% (95% CI, 8.7 to 17.6; *p* < 0.001). Grade 3 or 4 AEs occurred in 42.4% of patients with durvalumab and in 43.2% with placebo. In a preplanned subgroup analysis of patients with an EGFR mutation (n = 51) there was no clear evidence of clinical benefit with the use of durvalumab. There were seven and two deaths in the durvalumab and placebo arms, respectively, assessed by the investigator as possibly related to any systemic trial treatment, and included interstitial lung disease (ILD, in two patients); immune-mediated lung disease, pneumonitis, hemoptysis, myocarditis, and decreased appetite (one patient each) in the durvalumab group; and pneumonia and infection (one patient each) in the placebo group. Immune-mediated pneumonitis of any grade was reported in 3.7% vs. 1.8% of patients in the durvalumab and placebo group; grade 3 or 4 immune-mediated pneumonitis was reported in 1.2% and 1.0%, respectively. Interestingly, the subgroup analysis showed that the pCR rates in the durvalumab arm were higher in stage II vs. stage III (21.2% vs. 18.5% vs. 10.2% in stage II, IIIA and IIIB), while the EFS benefit was relatively higher in patients with stage IIIA disease [25].

### 3.5. KEYNOTE 671

Keynote 671 was a phase 3 study that evaluated a perioperative approach with pembrolizumab. Patients with resectable stage II, IIIA, or IIIB (N2 stage) NSCLC were randomized (n = 797) to either receive NA pembrolizumab or placebo, along with cisplatin-based chemotherapy for four cycles, followed by surgery and adjuvant pembrolizumab or placebo for a year. Notably, the dual primary endpoints were EFS and OS, while pCR was a secondary endpoint along with MPR and safety. A recent update with second interim analysis showed improvement in median OS (NR vs. 52.4 months, HR, 0.72; 95% CI, 0.56–0.93; *p* = 0.005). The 36-month OS rates were 71.3% vs. 64.0%. Median EFS was 47.2 vs. 18.3 months; and the 36-month EFS rates were 54.3% vs. 35.4%. The pCR rates were 18.1% vs. 4% in the experimental and placebo arms, respectively, while MPR was seen in 30.2% vs. 11%. Totals of 40.7% of the participants in the experimental arm and 36.6% of those in the placebo arm had TRAEs of grade 3 or higher in the NA phase, respectively. There were 1.0% and 0.8% of patients who had grade 5 events (from immune-mediated lung disease, pneumonia, and sudden cardiac death in one participant each during the NA–surgery phase and from atrial fibrillation in one patient during the adjuvant phase, in the pembrolizumab arm) in the experimental and control arms, respectively. There were 82.1% and 79.4% of patients who underwent surgery in the experimental and control arms, respectively. EFS-benefit with pembrolizumab was noted irrespective of pCR or MPR [13]. There were six deaths within 30 days of surgery in the pembrolizumab arm compared to two in the placebo arm. On 16 October 2023, FDA approved this perioperative regimen [31].

### 3.6. Neotorch

This was a randomized, double-blind, placebo-controlled, phase III trial to evaluate toripalimab (an anti-PD1 ICI) plus chemotherapy (three cycles before surgery and one after) followed by toripalimab maintenance for a year vs. chemotherapy alone in resectable stage II/III NSCLC, without EGFR/ALK alterations. A planned interim analysis for EFS of stage III subjects (n = 404) showed improved EFS in the toripalimab arm (NR vs. 15.1 months, HR, 0.40; 95% CI, 0.277–0.565; *p* < 0.0001), and crossed the pre-specified efficacy boundary. The MPR and pCR rates were better in the toripalimab arm at 48.5% vs. 8.4% and 24.8% vs. 1.0%, respectively. The incidence of grade 3 or more TRAEs was 63.4% vs. 54.0% and fatal AEs related to toripalimab/placebo were 0.5% vs. 0%. IrAEs were more frequent in the toripalimab arm, at 42.1% vs. 22.8% [32,33].

### 3.7. CheckMate 77T

This was a phase 3, double-blind RCT testing the efficacy of perioperative nivolumab in patients (n = 461) with stage IIA-IIIB without EGFR and ALK mutation/alteration (N2 included, AJCC 8th edition). The primary endpoint was EFS. With a median follow-up of 15.7 months, median EFS was better in in the nivolumab arm (NR vs. 18.4 months; HR, 0.58 (0.42–0.81); *p* = 0.00025). The pCR rates were 25.3% vs. 4.7% (OR, 6.64; 95% CI, 3.40–12.97) and the MPR rates were 35.4% vs. 12.1% (OR, 4.01; 95% CI, 2.48–6.49) in the experimental and control arms, respectively. The rates of definitive surgery were similar at 78% vs. 77%. Grade 3–4 TRAEs occurred in 32% and 25%, while surgery-related AEs were 12% and 12% in the experimental and control arms, respectively [26].

### 3.8. RATIONALE-315

This phase 3 clinical trial assessed the efficacy and safety of NA tislelizumab (an anti-PD1 monoclonal antibody) in patients with resectable stage II-IIIA NSCLC (n = 452), with EGFR and ALK alterations excluded. Patients received 3–4 cycles of tislelizumab or placebo with platinum-doublet chemotherapy followed by surgery and then eight cycles of adjuvant tislelizumab (every six weeks). The primary endpoint (MPR rates) were 56.2% vs. 15% (OR 7.5; 95% CI, 4.8–11.8; *p* < 0.0001) and the pCR rates were 40.7% vs. 5.7% (OR 11.5; 95% CI 6.2–21.5; *p* < 0.001) in the experimental and control arms, respectively, in the ITT population. The median follow-up was 16.8 months. In total, 15.9% vs. 23.8% of patients did not undergo surgery. Grade 3 or higher events were observed in 69.5% and 65.5% of the experimental and control arms, respectively [34].

**Table 2 cancers-16-01302-t002:** Trials with neoadjuvant ICI-chemo in stage III resectable NSCLC (design and outcomes).

Study	Population/ Staging Guideline	Experimental Arm	Control Arm	PEP	Stage 3 (n, Exp and Control Arm)	EFS/DFS in Stage III	pCR/MPR/ORR in Stage III	pCR by Stage (Exp vs. Control, %)	Adverse Events
CheckMate 816 Phase 3 Randomized [10]	IB-IIIA AJCC 7th	Three cycles of nivolumab plus chemotherapy q21d	Three cycles of platinum-doublet chemotherapy q21d	EFS, pCR	Stage IIIA 228 Exp—113 Control—115	Median EFS 31.6 vs. 15.7 months HR 0.54 (95% CI 0.37–0.80)	23% vs. 0%	21.8% vs. 0%	Grade ¾ TRAEs—33.5% vs. 36.9%
NADIM II Phase 2 Randomized [9]	IIIA-IIIB, excluded N3	Three cycles of nivolumab plus chemotherapy q21d; adjuvant nivolumab for 6 months	Three cycles of chemotherapy (carboplatin (AUC5) and paclitaxel (200mg/m^2^)) q21d	pCR	IIIA—77% vs. 83% IIIB—23 vs. 17%	2-year PFS—67.2% vs. 40.9 2-year OS—85% vs. 63.6%	pCR—37% vs. 7% MPR—53% vs. 14% ORR—75% vs. 48%	N2 single—42.1 vs. 0% N2 multiple—36.4 vs. 10% Initial N2—39 vs. 6.2%	Any grade TRAEs– 88% vs. 90% Grade 3–4 TRAEs—19% vs. 10%
NADIM Phase 2 Single-arm [35]	IIIA AJCC 7th	Three cycles of nivolumab plus chemotherapy (carboplatin (AUC5) and paclitaxel (200mg/m^2^)) q21d; adjuvant nivolumab for 6 months	NA	PFS at 24 months	IIIA—46	2-year PFS 77.1% 2-year OS—89.9%	pCR—56.5% MPR—73.9% ORR—76.1%	NA	Any grade TRAEs—93% Grade 3–4 TRAEs—34%
AEGEAN Phase 2 Randomized [16]	IIA-IIIB (N2) AJCC 8th	Four cycles of durvalumab plus chemotherapy q21d; adjuvant durvalumab (12 cycles)	Four cycles of placebo plus platinum-doublet q21d; adjuvant placebo	pCR, EFS	IIIA (173 and 165) IIIB (88 and 98)	Median EFS IIIA—NR vs. 19.5 months (HR 0.57, 95% CI, 0.39–0.83) IIIB—31.9 vs. 18.9 months (HR 0.83, 95% CI, 0.52–1.32)	NA	pCR IIIA—18.5% vs. 4.8% IIIB—10.2% vs. 3.1%	Any grade IRAEs—23.7% vs. 9.3% Grade ¾ IRAEs—4.2% vs. 2.5%
TD-FOREKNOW Phase 2 [30] Randomized [30]	IIIA IIIB (T3N2 only) AJCC 8th	Three cycles of camrelizumab plus chemotherapy q21d	Three cycles of chemotherapy alone (nab-paclitaxel (130 mg/m^2^ d1/d8) plus platinum (cis/carbo/nedaplatin))	pCR	IIIA (39 and 36) IIIB (13 and 9)	Median EFS—NR, HR 0.52 (95% CI, 0.21–1.29) 2-year EFS—76.9% vs. 67.6% Median DFS—NR, HR 0.54 (95% CI, 0.19–1.54)	pCR—32.6% vs. 8.9% MPR—65.1% vs. 15.6% ORR—72.1% vs. 53.3% CR—25.6% vs. 8.9%	IIIA—30% vs. 8.3% IIIB—38.5% vs. 11%	Any grade IRAEs—53.5% (all were grade 1 or 2) vs. 0%
KEYNOTE 671 Phase 3 Randomized [13]	II-IIIB (N2) AJCC 8th	Four cycles of neoadjuvant pembrolizumab plus chemotherapy q21d; adjuvant pembrolizumab (13 cycles)	Four cycles of placebo plus chemotherapy (cisplatin plus gemcitabine or pemetrexed) q21d; adjuvant placebo	EFS and OS	IIIA—217 vs. 225 IIB—62 vs. 54	HR for EFS—0.54 (0.42–0.70)	NA	NA	Any grade IRAEs—25.3% vs. 10.5%
Neotorch Phase 3 Randomized [32,33]	II-IIIB (N2) AJCC 8th	Three cycles of neoadjuvant toripalimab plus chemotherapy q21d; adjuvant toripalimab (13 cycles)	Three cycles of placebo plus platinum-doublet chemotherapy q21d; adjuvant placebo	MPR and EFS in stage III	202 in each arm	Median EFS —NR vs. 15.1 months (HR, 0.40, 95% CI, 0.28–0.57)	pCR—24.8% vs. 1.0%, MPR—48.5% vs. 8.4%	NA	Grade 3–4 TRAEs—63.4% vs. 54%; Any grade IRAEs—42.1% vs. 22.8%
CheckMate 77T Phase 3 Randomized [26]	IIA-IIIB (N2) AJCC 8th	Four cycles of neoadjuvant nivolumab plus chemotherapy q21d; adjuvant nivolumab (1 year)	Four cycles of placebo plus chemotherapy q21d; adjuvant placebo	EFS	NA	Median EFS stage IIA-IIIB)—NR vs. 18.4 months	pCR (stage IIA-IIIB)—25.3% vs. 4.7% MPR—35.4% vs. 12.1%	NA	Grade 3–4 TRAEs—32% vs. 25%
RATIONALE-315 [34]	II-IIIA AJCC 8th	Three–Four cycles of neoadjuvant tislelizumab plus chemotherapy q21d; adjuvant tislelizumab (q6w, up to 8 cycles)	Three–Four cycles of placebo plus platinum- doublet chemotherapy q21d; adjuvant placebo	MPR and EFS	NA	NA	NA	NA	Grade 3–4 TRAEs—69.5% vs. 65.5%

DLTs—dose limiting toxicities; ORR—overall response rate; CNS—central nervous system; DCR—disease control rate; PFS—progression free survival; OS—overall survival; TN—triple negative; mets—metastasis; ICC—intracranial control rate; TTSP—time to systemic progression; AE—adverse events; CR—complete response; PR—partial response; SD—stable disease; CBR—clinical benefit rate; IRAEs—Immune-related adverse events; RT—radiation therapy; Pts—patients; pCR—pathological complete response; EFS—event free survival; NA—not available.

**Table 3 cancers-16-01302-t003:** Surgical outcomes in trials with neoadjuvant chemo-immunotherapy in stage III resectable NSCLC.

Study	Median Time to Surgery	Underwent Surgery (Exp vs. Control, %)	Delay in Surgery (Exp vs. Control, %)	VATS; Thoracotomy; VATS to Thoracotomy	Pneumonectomy	R0 (% of Underwent Surgery)	Operative Time (min)	Length of Hospital Stay (Days)	Downstaging	Surgery-Related AEs
CheckMate 816 [10]	5.3 vs. 5 weeks	83.2% vs. 72.2%	23.4% vs. 13.3%	28% vs. 16%; 59.6% vs. 60.2%; 10.6% vs. 20.5%	17% vs. 30.1%	83% vs. 78.3%	185.5 vs. 218	9 vs. 10	30.7% vs. 23.5% (all stages)	41.6% vs. 46.7%
NADIM II [9]	7 weeks in both arms	93% vs. 69%	Median time to surgery > 7 weeks 32% vs. 35%	-	11.3% vs. 10%	94.3 vs. 85%	-	-	Nodal downstaging 72% vs. 40%	41.5% vs. 35%
NADIM [35]	-	89%	-	-	-	-	-	-	90% had pathological downstaging	46.3%
AEGEAN [16]	4.8 weeks in both arms	77.5% vs. 76.6%	14.5% vs. 16.8% ^#^	NA	7.4% vs. 7.8%	94.7% vs. 91.3%	NA	NA	NA	NA
TD-FOREKNOW [30]	4.7 vs. 4.6 weeks	93% vs. 93.3%	Median time to surgery > 6 weeks in 12.5% vs. 2.4%	55% vs. 61.9%; 37.5% vs. 33.3%; 7.5% vs. 4.8%	10% vs. 19%	92.5% vs. 85.7%	151.5 vs. 150.0	10 vs. 9	53.5% vs. 44.4%	40% vs. 33.3%
KEYNOTE 671 [13]	NA	82.1% vs. 79.4%	NA	NA	11.4% vs. 12.3%		NA	NA	NA	Any grade, 71.1% vs. 71.3% Grade 3–5, 25.8% vs. 21.5% Grade 5, 2.8% vs. 1.6%
Neotorch [32,33]	NA	82.2% vs. 73.3%	NA	NA	9% vs. 9.5%	95.8% vs. 92.6%	NA	NA	NA	NA
CheckMate 77T [26]	NA	78% vs. 77%	NA	NA	NA	89 vs. 90%	NA	NA	NA	12% vs. 12%
RATIONALE-315 [34]	NA	84.1% vs. 76.2%	NA	NA	NA	NA	NA	NA	NA	NA

DLTs—dose limiting toxicities; ORR—overall response rate; CNS—central nervous system; DCR—disease control rate; PFS—progression free survival; OS—overall survival; TN—triple negative; mets—metastasis; ICC—intracranial control rate; TTSP—time to systemic progression; AE—adverse events; CR—complete response; PR—partial response; SD—stable disease; CBR—clinical benefit rate; TRAEs—treatment-related adverse events; IRAE—immune-related adverse events; RT—radiation therapy; Pts—patients; pCR—pathological complete response; EFS—event free survival; NA—not available; ^#^ A surgical delay is defined as surgery occurring more than 40 days after the last dose of study treatment in the neoadjuvant period.

## 4. Discussion

### 4.1. Patient Selection for Neoadjuvant ICI-Chemotherapy

An NA approach provides the opportunity for prompt treatment initiation, better compliance, and pathological assessment of treatment effect, which can be used to guide adjuvant treatment and eradicate micro-metastatic disease [36]. The cons of this strategy include the delays in time to surgery, the low-probability risk of progression or toxicity to systemic treatment which may preclude curative surgery, and an underestimation of the true stage at the time of pathological assessment. Nevertheless, there is consensus on the benefit of NA ICI-chemo in resectable NSCLC. In all trials, more patients in the experimental arms had surgery compared to those in the control arms. Additionally, an argument can be made that disease progression on systemic therapy is an indicator for more aggressive disease biology and that these patients would likely have not benefited from surgery. This phenomenon can also be seen in purely adjuvant trials of chemotherapy and ICI; for example, in IMpower010, among patients with resected NSCLC who received adjuvant therapy, 18% had disease progression before completing adjuvant ICI [37,38]. More serious harm may arise in a scenario where a patient may have severe toxicity, either from chemotherapy or ICI, precluding or delaying surgery, or even resulting in death. As such, careful patient selection is vital.

Unfortunately, there is not enough evidence that may help in choosing patients with resectable NSCLC for NA ICI-chemo and surgery vs. immediate surgery and adjuvant chemotherapy and ICI. The benefits with the NA approach are higher in patients with higher stage disease, as these patients have a higher probability of recurrence accompanied by worse prognosis, especially as the response and survival benefits with NA ICI-chemo seem to be independent of stage in most trials. Nonetheless, in Keynote 671, the EFS benefit was greater in patients with stage 2 than in those with stage 3 disease [13].

The role of PD-L1 TPS in predicting the likelihood of response does exist, with patients with PD-L1 TPS scores ≥ 50% having a higher pCR rate compared to those having PD-L1 TPS 1–49% or <1%. In NADIM 2, the predictive value (AUC) of PD-L1 TPS for pCR was 0.72 (95% CI 0.58–0.87) [9]. Although, a PD-L1 TPS may enable us to gauge the clinical benefit and risk-ratio to help in making an informed decision in the clinic, it does not help in selecting patients, as even patients with PD-L1 TPS < 1% may have a pCR.

The response rate and survival benefits in patients with an oncogene driven NSCLC, especially those with EGFR, ALK, ROS1, RET and NTRK fusion, are low [39,40,41]. Most trials assessing ICIs in resectable NSCLC excluded patients with EGFR and ALK mutations or alteration/rearrangement. One trial showed a lack of benefit with NA ICI-chemo in patients with EGFR alterations, although the numbers were small (n = 51) [25]. In the NADIM trial, patients having mutations that confer resistance to ICIs, such as STK11, KEAP1, RB1, and EGFR, had poor outcomes. In fact, among the four patients with an STK11 or KEAP mutation in the experimental arm, two patients did not undergo surgery while the other two had an incomplete response [35]. With current data from these NA trials, there is no evidence to exclude any group of patients other than those with EGFR/ALK alterations. It is questionable whether patients with other targetable/driver mutations, such as ROS1 or RET, should be treated with NA ICI-chemo, based on data showing the limited efficacy of immunotherapy in patients with advanced lung cancer who have these mutations [42]. Data on response and survival on patients with these alterations may shed more light [43]. It should be kept in mind that targeted therapy after ICIs have been associated with AEs in patients with targets such as RET [44,45]. Patients with mutations such as KRAS, BRAF and c-MET may benefit from NA ICI-chemo based on the extrapolation of the benefit of immunotherapy in patients with advanced lung cancer who have these mutations [42].

### 4.2. Patient Selection for Adjuvant Immunotherapy

There is a lack of data regarding patient selection and the absolute benefit of adjuvant ICI after NA ICI-chemo. In a post hoc analysis, the NADIM II trial showed that among patients who had an R0 resection, those who completed adjuvant ICI had a better survival outcome than those who did not complete adjuvant ICI, even after excluding patients with pCR [9]. Exploratory analysis revealed an EFS benefit in patients receiving adjuvant ICI in the KEYNOTE 671 trial [13]. There appears to be a widening of survival curves in KEYNOTE 671 in those without a pCR (ICI vs. placebo), while in CheckMate 816 the benefit seems to level off earlier [10,13,27]. The first question that arises is should patients with a pCR receive adjuvant ICI, given that they have excellent prognoses and adjuvant treatment might subject them to irAEs? Secondly, do patients without a pCR or MPR benefit with adjuvant ICI, and should these patients with non-pCR receive more of the same ICI that failed to eradicate their disease in the first place? Should patients without a pCR have escalated treatment with additional ICIs (CTLA4 or LAG-3 inhibitors) or ADCs to improve outcomes? At this time there is no evidence on whether the response to NA therapy should guide whether adjuvant therapy is offered or not. Randomized data from trials specifically designed to address these issues are needed to understand the extent of the absolute benefit as well as to come to a consensus on optimal patient selection, timing and duration (6 months vs. 12 months or longer) of adjuvant ICI after NA treatment.

### 4.3. Choice of Immunotherapy

With many trials meeting their primary endpoints and showing an improvement in pCR and survival, various options are available and likely to be approved as survival data accumulate further. The regimens used in CheckMate 816 and Keynote 671 are approved by the FDA [31,46]. Cross-trial comparison and subgroup analysis should be conducted with caution, as there were differences in trial design (open-label vs. blinded), stage (e.g., AJCC version 7th or 8th), nodal burden, patient population, chemotherapy backbone, method of pCR assessment (IASLC criteria vs. immune-related pathological response criteria (irPRC) criteria), etc. [47,48,49]. At the same time there are differences with regards to strategy in different trials—whether NA or peri-operative. The additional impact of adjuvant ICI on survival is unknown; some trials have shown a trend towards benefit in subsets of patients without a pCR, although confirmatory evidence is lacking in the absence of randomized data [9,13]. The limited and short treatment duration of only NA ICIs, with comparable efficacy, seems more feasible economically. That said, such decisions should be based on informed and shared decision-making. As with efficacy, the safety of NA ICIs has also been comparable and similar across trials, with no new safety signals. Additionally, OS data would be important, because ongoing ICI could have implications for the efficacy of ICI in treating recurrent disease. Head-to-head trials of the different ICIs are not planned by industry and are unlikely to happen unless academic groups undertake them. It could also be debated whether the existing trial database (as reviewed in this article and elsewhere) provides enough justification to conduct large head-to-head trials of the different ICIs.

### 4.4. Choice of Chemotherapy Backbone

Various trials have used different chemotherapy backbones, although platinum-doublet remains the common denominator. The choice of platinum (cisplatin vs. carboplatin) may be affected by host factors such as age, hearing acuity, renal functions, cardiovascular status, performance status, pre-existing neuropathy and other comorbidities, like diabetes mellitus. NADIM II used carboplatin (area under the concentration–time curve (AUC) 5 mg per milliliter per minute) and paclitaxel (200 mg per square meter), whereas NADIM used an AUC 6 dose for carboplatin [9,35]. AEGEAN allowed for more options for platinum-doublets (dosing or partner drug) as per the discretion of the investigator [25]. TD-FOREKNOW used platinum and nab-paclitaxel so as not to use steroids with ICI [30]. Patients with squamous histology in KEYNOTE 671 were treated with cisplatin/and gemcitabine while many trials mandated carboplatin and paclitaxel for this population [13]. In CheckMate 77T and Neotorch, a taxane (docetaxel or paclitaxel) was mandated in patients with squamous histology [26,33]. In the absence of any clear evidence of a superior specific partner to platinum, the choice should remain open and be based on host factors, histology, and shared decision-making. For well-differentiated adenocarcinomas, the more tolerable platinum-pemetrexed may be a preferable choice. For poorly differentiated tumors and squamous histology, platinum-paclitaxel may be preferable. The choice of chemotherapy may be more critical for patients with PD-L1 negative/weak-positive tumors in which the relative role of chemotherapy on response may be more pronounced.

Before the advent of ICIs in early-stage NSCLC, the standard of care for resectable NSCLC was upfront surgery followed by four cycles of adjuvant chemotherapy for most of the eligible patients. NA chemotherapy plus/minus radiation was not widely used as there was no added benefit to survival with NA chemotherapy, compared to adjuvant chemotherapy, and there is a risk of losing an opportunity to resect the tumor if there is progression or toxicity on systemic treatment. However, it should be acknowledged that in the real world only less than half of eligible patients receive adjuvant chemotherapy after surgery, with estimates as low as 27% is one study [50,51,52]. In the JBR.10 and ANITA trials up to 40–50% of patients did not complete the planned course of treatment [53,54]. However, the control arm in the NA/peri-operative trials has been NA chemotherapy rather than adjuvant chemotherapy. The reasons for this approach are to achieve a more comparable and pragmatic control arm and the lack of any evidence of inferiority of NA chemotherapy compared to adjuvant chemotherapy.

The optimum number of NA cycles (three vs. four) is also debatable, and although the historical convention has been to use four cycles, some trials have used three cycles. From the evidence of CheckMate 816, three cycles seemed to be associated with clinically meaningful response; however, it is unknown, if four cycles (as in KEYNOTE 671) are better. It is noteworthy that the pCR rates in CheckMate 816 (three cycles) and 77T (four cycles) were very similar [10,13,26]. Whatever the choice of cycle number, the patients should be carefully watched clinically so that any progression or toxicity can be caught early, at a time when surgery or other curative treatment is still possible.

### 4.5. Assessment of Response

In most trials, the radiographic response to NA ICI-chemo was evaluated with RECIST, version 1.1. Patients had a positron emission tomography–computed tomography (PET–CT) scan or CT scan before surgery. The possibility of pseudo-progression and reactive FDG uptake should be considered when interpreting responses on imaging. The availability of PET–CT may be limited at cancer centers. Serial chest X-rays after each cycle may help to re-evaluate the primary mass to rule out obvious progression during treatment.

The assessment of pathological responses outside of trials should also be stringent and based on guidelines [47,55]. Some trials such as CheckMate 816 used the immune-related pathological response criteria (irPRC), while other trials like AEGEAN used the criteria outlined by IALSC, with some minor differences between the two sets of criteria [47,48,49]. It is clear that the degree of response affects the outcomes, and there should be a practical standard set for pathology departments to assess residual tumors. Considering that patients with pCR have much improved outcomes compared with the subset of those having MPR, those who have an MPR and not a pCR should be considered differently. Also, the percentage of residual viable tumor (RVT) predicted the EFS for nivolumab plus chemotherapy in the CheckMate 816 trial (area under the curve, 0.74) with the 2-year EFS rates being 90%, 60%, 57%, and 39% for patients with 0–5%, >5–30%, >30–80%, and >80% RVT, respectively [56]. Biomarkers such as ctDNA may also help in predicting a response and aid clinical decision-making for issues such as switching, escalating or de-escalating treatment strategies after surgery.

### 4.6. Toxicity and Quality of Life

As elaborated in Table 2, the rates of TRAEs were similar in the experimental and control arms, although, as expected, there were more irAEs in patients receiving ICIs. These irAEs were manageable and the occurrence of irAEs that may preclude a surgery was very low. In view of the potentially detrimental effects of irAEs such as pneumonitis on pulmonary reserves and suitability of surgery, this infrequent but possible side effect should be discussed to ensure informed decision-making. Patients should be educated regarding the signs and symptoms that may aid in prompt recognition. Patients should be prescreened for any risk factor that may increase the risk of irAEs, for example ILD. In case of pneumonitis, prompt initiation of individualized and guideline-based management may help to abate the damage to pulmonary function [57]. In such patients, pulmonary function tests should be repeated before surgery to estimate the pulmonary reserve. Long-term data regarding chronic irAEs are still awaited and require more consideration in patients who are treated with adjuvant ICIs [58].

Systemic treatment can decrease tumor burden, resulting in prompt symptom improvement in patients with a response. Some patients may experience AEs which may be associated with a decrement in quality of life (QoL). Most of the trials discussed above have looked at health-related quality of life (HRQoL) outcomes as an exploratory endpoint. In the CheckMate 816 trial, no detrimental effect was noted by addition of ICI to chemotherapy and most patients reported “no problems” for individual EQ-5D-3L dimensions at baseline and during neoadjuvant treatment [59]. Such data on HRQoL are awaited from other trials, especially in the post-surgery period, as well as information on the effect of adjuvant immunotherapy on quality of life.

### 4.7. Salvage Treatment

Guidelines are lacking regarding the optimal management of patients who are not able to undergo definitive surgery due to progression of the disease, finding of unresectable disease at the time for surgery, or toxicity precluding surgery. For localized disease, CTRT may still be an option, although the risk of pneumonitis during radiation may be higher if treatment is given in close proximity to the last dose of ICI. For patients with more advanced disease, standard treatment for stage IV disease based on PD-L1 and biomarker status may be appropriate. However, situations may arise where there is obvious progression on NA ICI with platinum-doublet chemotherapy. In such a scenario, there is a lack of evidence on whether first-line ICI-chemo regimens for advanced NSCLC would still be an option. The progression of disease in this setting would mean aggressive biology and poor prognosis. Other options include intensifying treatment by adding a CTLA4 inhibitor or next line single-agent chemotherapy (Figure 2).

### 4.8. Surgical Perspective and Downstaging of Unresectable Disease

The nodal status (especially N2) should be carefully assessed and pathologically confirmed before and after NA systemic treatment. The window for timing of surgery after NA ICI-chemo is important. In trials, the median time to surgery was 5–7 weeks after the last cycle. A delay might lead to amplified infiltration by immune cells manifesting as fibrosis, which may make the surgery difficult. It might also lead to progression in the absence of ongoing treatment. Notably, more patients had surgery after NA ICI-chemo compared to chemotherapy alone in CheckMate 816 trial and duration of surgery was shorter in the experimental arm [10]. Similar observations have been made in other trials. A recent study reported no difference in pathological response between the experimental and control groups, with the median time to surgery after the last ICI dose being 13 and 22 days, respectively, with the authors concluding that patients may undergo resection as early as 2 weeks after the last dose [60]. None of the trials reported a new safety signal, such as increased surgical complications or longer duration of surgery, although even subclinical pneumonitis may impact pulmonary reserve and may have a bearing on cardio-pulmonary recovery after surgery.

Historically, NA strategy with chemotherapy has only been unsuccessful in downstaging unresectable disease. For example, in the surgical arm of the EORTC 08941 trial, only 50% of patients had an R0 resection and 41% of patients had pathological down-staging when patients with N2 disease were treated with NA chemotherapy (three cycles) followed by surgery and there was significant morbidity and mortality (8% underwent re-operation and 4% had operative 30-day mortality [61]). The eligibility of NA ICI-chemo trials mandated that fit patients had upfront resectable disease. At the same time, the boundaries of resectability are being pushed considering the high response rate to NA ICI-chemo. The question of the extent of resection needed after a radiographic response is still unanswered. For patients who are borderline resectable due to the anatomical location of a tumor, it is unclear whether radiographic downstaging leads to surgical downstaging. For example, tumors involving the hilum commonly require a pneumonectomy to achieve an R0 resection. However, radiographic regression does not always correspond to improved access to the hilum. Intraoperatively, fibrosis after NA therapy results in challenging dissection planes that make it difficult to discern fibrosis from residual malignancy. As such, in many instances, a pneumonectomy may still be required despite radiographic downstaging. Thus, it is currently recommended that only resectable NSCLC be treated with NA ICI-chemo followed by surgery. For more advanced diseases such as bulky N2 or N3, or any situation where the MDT feels that the disease is not resectable upfront, the standard should be to treat with ‘definitive’ CTRT followed by consolidation ICI, based on data from the PACIFIC study [5]. Notably, the PACIFIC trial had 53% of patients with unresectable stage IIIA disease [62]. Even though the NA or peri-operative data suggest downstaging irrespective of nodal status, none of the trials included upfront unresectable patients. The interest in downstaging borderline resectable and unresectable disease has increased, as the response rates are significantly better by adding ICI to chemotherapy, which can make tumors amenable to R0 resection, although this approach has not been tested against the standard concurrent CTRT followed by durvalumab. Diligent caution should be exercised in using this approach if the intent is to elicit a response to allow a lobectomy after NA ICI-chemo; for example, if the patient requires a pneumonectomy upfront but may not tolerate pneumonectomy. Ongoing trials are evaluating NA-ICI in more advanced disease [63,64,65,66,67].

## 5. Future

Two interesting arenas of ongoing research include, firstly, improving the rate of pCR and survival outcomes and, secondly, predictive biomarkers and models.

### 5.1. Strategies to Improve Outcomes

There are several ongoing trials testing strategies to improve the rates of pCR and other outcomes (Table 4). These trials are evaluating if the addition of ICIs with a different checkpoint blockade (e.g., CTLA4 or LAG-3 inhibitors) or other modalities like radiation can improve the outcomes, especially in patients with low PD-L1 expression. For example, the INCREASE trial, a single-arm, prospective, phase II trial, evaluated the efficacy and safety of adding NA nivolumab-ipilimumab (NIVO/IPI) to standard induction CTRT in patients with either resectable, or borderline resectable, T3-4 N0–1 NSCLC. Patients received NIVO/IPI on day 1 of CRT (50–60 Gy in 25 fractions) with a platinum-doublet, followed by nivolumab after three weeks and had surgery six weeks after the last day of radiotherapy [68]. The interim results showed impressive MPR and pCR rates of 79% (n = 19/24) and 63% (n = 15/24), respectively. The strategy was safe and did not interfere with the anatomical resection. One postoperative death occurred on day-96 due to immune-related pneumonitis, although no 30- and 90-day mortality was observed [69]. SAKK 16/18, a single-arm phase II trial in patients with stage IIIA-B (N2), used a similar strategy and assessed concurrent RT with ICI after chemotherapy. This was followed by surgery and adjuvant ICI. Interim results showed the safety of this approach [70]. Although the elevation in the pCR rate associated with intensification of the regimen might be impressive in the INCREASE trial, it is not clear whether the addition of ipilimumab or RT is responsible, or both; moreover, if due to the RT, it is not obvious whether that increment necessarily carries the same long-term benefit in freedom from (out-of-field) distant metastatic disease as it does if induced purely by the addition of systemically-administered ICI to the chemotherapy.

### 5.2. Predictive Biomarkers and Models

ctDNA is a promising biomarker for response in early-stage solid tumors [71]. Recently, Gouda et al. proposed the Liquid Biopsy Response Evaluation Criteria (LB-RECIST) to be used in the context of clinical trials [72]. The newer platforms have increased sensitivity and specificity [73]. The potential role of ctDNA in resectable NSCLC includes diagnosis, prognosis, predicting response (pCR or MPR) leading to treatment escalation/de-escalation, predicting need for adjuvant treatment, and predicting a recurrence after surgery (Figure 3).

Ongoing studies are looking at biomarkers to predict a response [74,75,76,77]. In CheckMate 816, 46% (11/24) patients achieving a ctDNA clearance before cycle 3 in the experimental arm had a pCR compared to no pCR in patients with detectable ctDNA [10]. In the NADIM trial, the pre-treatment levels of ctDNA were predictive of PFS and OS [78]. In the AEGEAN trial, all patients achieving pCR had ctDNA clearance by cycle four on day 1, and ctDNA response, as well as clearance, was observed as early as cycle two on day 1 [79]. Similar findings were noted in the phase II CTONG1804 trial, as patients with ctDNA negativity after NA ICI-chemo were more likely to have a pCR and better 18-month EFS rate compared to patients with detectable ctDNA [80]. Ferrer at al. aimed to develop a machine learning algorithm to predict pCR status based on multimodal baseline data, and looked at patient and tumor characteristics as well as radiomics in a cohort of 28 patients from the NADIM trial. A model including the neutrophil to lymphocyte ratio, mutations, histology, and radiomics achieved an AUC of 0.76 in predicting pCR [81].

Most of the studies have looked at tumor parameters (e.g., PD-L1, TMB, mismatch repair deficiency, and microsatellite instability) as predictive biomarkers. Should the health of the host/patients’ immune system not be scrutinized more carefully when selecting patients for ICIs? There is evidence that host factors such as circulating immune compartments, microbiomes, body composition, and germline genetic factors modulate response, as well as immune toxicity to immunotherapy [82,83,84]. Pre-PLaN (NCT06250829) is an exploratory translational study assessing host (fecal microbiome, immune cell subsets, and body composition) and tumor variables (PD-L1, TMB, radiomics, tumor microenvironment, in-depth tumor genetic analysis, and ctDNA), to predict response and toxicity to NA ICI-chemo in resectable NSCLC [85]. Similarly, the PROPHETIC (NCT05736029) study is an observational study using multi-omics to predict a response to ICIs (ctDNA, epigenetic patterns, microbiome profile, and proteomic profile) [86].

Most of the data for biomarkers in neoadjuvant setting are from exploratory analysis of translational data. Future trials should be designed with interventions and treatment decisions based on dynamic ctDNA monitoring to generate evidence for clinical utility. If a biomarker or model is able to reliably predict a pCR, it would be interesting to see if these patients would still need a resection or whether they can they be spared from surgery.

**Table 4 cancers-16-01302-t004:** Ongoing trials with neoadjuvant immunotherapy in NSCLC.

Study	Population (Stage)	Experimental Arm	Control Arm	Primary Endpoint
NCT04973293 [87] Single-arm	II-IIIA	Four cycles of sintilimab, bevacizumab, carboplatin and pemetrexed	None	Safety
NCT04875585 [88] Single-arm, phase II	IA3-IIIA	Pembrolizumab/lenvatinib followed by surgery followed by adjuvant pembrolizumab	None	MPR
NCT05684276 [89] Single-arm, phase II	IIB, IIIA and T3N2 (IIIB) Pancoast tumor	Three cycles of nivolumab, carboplatin and paclitaxel followed by surgery followed by adjuvant nivolumab	None	R0 resection rate
NCT06055465 [63] Single-arm	potentially resectable II-III	Four cycles of sacituzumab govitecan and pembrolizumab followed by surgery followed by adjuvant pembrolizumab	None	pCR
NCT04943029 [64] Single-arm	Unresectable III (IIIA-bulky N2, III B, IIIC)	Induction immuno-chemotherapy or immunotherapy prior to resection or definitive chemoradiotherapy	None	MPR
NCT04926584 [65] Single-arm	stage III-IVA (oligometastatic)	Induction immuno(chemo)therapy followed by surgery	None	patients completing definitive therapy
NCT04379739 [90] Single-arm	resectable II-III	Camrelizumab plus oral apatinib (two–four cycles) followed by surgery	None	MPR
NCT04245514 [91] Single-arm, phase II	resectable NSCLC	Neoadjuvant radiotherapy with durvalumab plus chemotherapy followed by surgery followed by adjuvant durvalumab	None	EFS at 12 months
NCT02572843 [92] Single-arm	IIIA (N2)	Three cycles cisplatin/docetaxel followed by two cycles of neoadjuvant immunotherapy with durvalumab	None	EFS at 12 months
NCT06109402 [93] Randomized (N = 160)	II-IIIB (N2)	Neoadjuvant immunochemotherapy followed by surgery followed by adjuvant immunotherapy	Surgery followed by adjuvant immunochemotherapy and immunotherapy	3-year EFS
NCT04699721 [94] Single-arm	IIIA and IIIB (T3N2)	Three cycles of nivolumab, nab-paclitaxel and carboplatin plus BiFico (oral probiotic)	None	Efficacy and safety
NCT05911308 [95] Single-arm	resectable NSCLC	Durvalumab plus platinum-doublet chemotherapy in combination with abequolixron (RGX-104), an LXR/ApoE agonist, followed by surgery	None	Safety and feasibility of surgery
NCT06065813 [96] Single-arm	IIA-IIIA	Neoadjuvant toripalimab with radiation (40–45 Gy) followed by surgery	None	MPR nad EFS
NCT03237377 [97] Single-arm	IIIA	Neoadjuvant durvalumab (3 doses) with radiation (45Gy in 25 fractions) followed by surgery and adjuvant chemotherapy	None	Safety and feasibility
NCT04202809 [98] Randomized (N = 90)	IIIA and IIIB	Chemo- and radiochemotherapy + durvalumab followed by surgery	Chemo- and radiochemotherapy followed by surgery	2-year PFS
NCT05798845 [99] Randomized phase II	II-IIIA	Radiation to primary tumor and toripalimab (two cycles) followed by surgery	Chemotherapy + toripalimab followed by surgery	pCR rate
NCT05319574 [100] Single-arm	II-IIIA	Neoadjuvant SBRT plus immunochemotherapy followed by surgery	None	MPR
NCT05940532 [66] Single-arm	Unresectable and Stage III	Three cycles of sugemalimab and chemotherapy followed by adjuvant nivolumab	None	ORR
NCT05766800 [67] Randomized (N = 100)	Unresectable, stage III treated with Serplulimab plus chemotherapy (4 cycles)	Surgery for downstaged and resectable tumors	RT for downstaged and resectable tumors	EFS
NCT05825625 [101] Single-arm phase II (n = 35)	II-IIIB (T3N2 only)	Chemotherapy in combination with atezolizumab and tiragolumab (2 cycles) followed by surgery	None	MPR
NCT04205552 [102]	IB-IIIA	Arm 1—Nivolumab (2 cycles) followed by surgery Arm 2—Nivolumab/Relatlimab (2 cycles) followed by surgery	None	Feasibility

pCR—pathological complete response; MPR—major pathological response; EFS—event free survival; ORR—overall response rate.

## 6. Conclusions

Recent evidence confirms the benefit of NA ICI-chemo in terms of pCR and survival in resectable NSCLC. Nevertheless, the trials are heterogenous with respect to the patient population, ICI used, number of cycles and timing/duration of treatment with adjuvant ICI. NA ICI-chemo should be discussed with all patients with resectable NSCLC as a standard option. Further data are needed regarding promising biomarkers with good clinical utility to enable patient selection for NA ICI-chemo strategy, as well as to select which patients would benefit from additional treatment after surgery. In addition to efficacy, toxicity should be taken into consideration when making clinical decisions. From the perspective of a healthcare system, the financial burden and cost-effectiveness of the treatment approach (NA vs. perioperative) needs careful consideration when deciding on approval. In the absence of definitive data for downstaging disease in borderline resectable patients, caution must be exercised before using NA ICI-chemo in this setting. Ongoing trials are evaluating strategies to further improve the outcomes in this patient population.

## Figures and Tables

**Figure 1 cancers-16-01302-f001:**
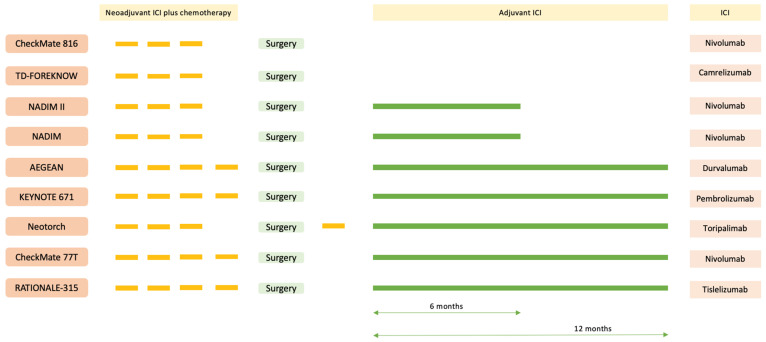
Schema of phase II/III trials with neoadjuvant immune checkpoint inhibitors (ICIs) in patients with resectable NSCLC.

**Figure 2 cancers-16-01302-f002:**
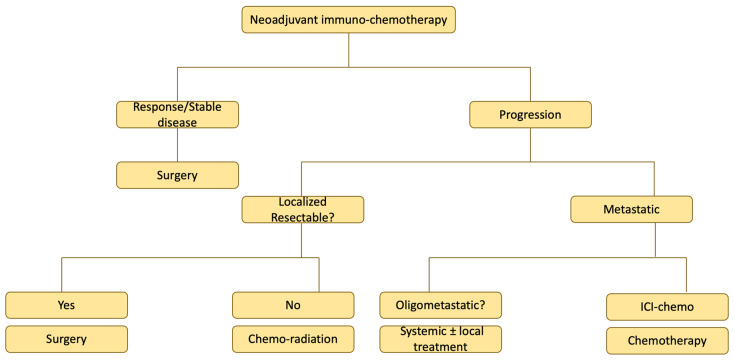
Potential options for salvage treatment.

**Figure 3 cancers-16-01302-f003:**
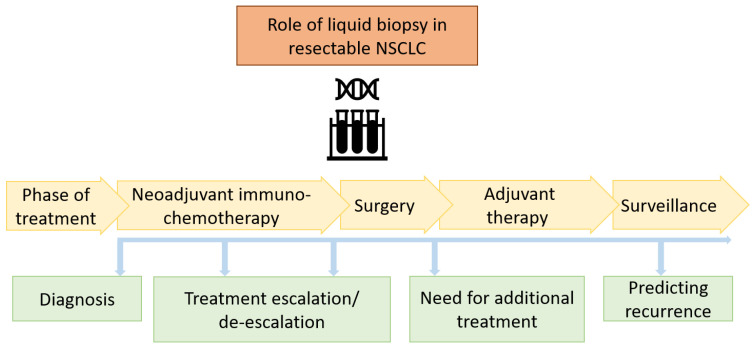
Potential role of ctDNA as a biomarker in resectable NSCLC in the era of immunotherapy.

**Table 1 cancers-16-01302-t001:** Comparison of 8th and proposed 9th edition of TNM staging for NSCLC.

		TNM 8th Edition				TNM 9th Edition (Proposed)				
T category	Subcategory	N0	N1	N2	N3	N0	N1	N2		N3
								N2a	N2b	
T1	T1a			IIIA	IIIB			IIB	IIIA	IIIB
	T1b			IIIA	IIIB			IIB	IIIA	IIIB
	T1c			IIIA	IIIB			IIB	IIIA	IIIB
T2	T2a			IIIB	IIIB			IIIA	IIIB	IIIB
	T2b			IIIB	IIIB			IIIA	IIIB	IIIB
T3	T3		IIIA	IIIB	IIIC		IIIA	IIIA	IIIB	IIIC
T4	T4	IIIA	IIIA	IIIB	IIIC	IIIA	IIIA	IIIB	IIIB	IIIC

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
