# Peer review of "An Updated Review of Management of Resectable Stage III NSCLC in the Era of Neoadjuvant Immunotherapy"

_cancers, 2024, doi:10.3390/cancers16071302_

Round 1
Reviewer 1 Report
Comments and Suggestions for Authors
The review is correct and provides summary information on the current status of neoadjuvant treatment with chemo and immunotherapy. The discussion, although it provides few solutions to the current controversies, is also interesting.
The phase II CTONG1804 trial could be added, which also provides information on ctDNA.
It would also be interesting to comment on whether radiotherapy would have a role in neoadjuvant treatment, especially in patients with low PD-L1 expression.
Author Response
Dear Reviewer
Thank you for reviewing the manuscript. Please find below the response to the comments.
Comment 1: The review is correct and provides summary information on the current status of neoadjuvant treatment with chemo and immunotherapy. The discussion, although it provides few solutions to the current controversies, is also interesting.
Response 1: Thank you for the positive feedback.
Comment 2: The phase II CTONG1804 trial could be added, which also provides information on ctDNA.
Response 2: Thank you for the suggestion. We have added the CTONG1804 trial (page 16, line 610-613).
Comment 3: It would also be interesting to comment on whether radiotherapy would have a role in neoadjuvant treatment, especially in patients with low PD-L1 expression.
Response 3: That is a very valid point. We have discussed role of radiation section 8a, including INCREASE and SAKK 16/18 trial. We have now added a comment to emphasize that this may be a strategy ‘especially in patients with low PD-L1 expression’ (page 16, line 579).
Reviewer 2 Report
Comments and Suggestions for Authors
This review provides a comprehensive and updated overview of the treatment of resectable Stage III non-small cell lung cancer (NSCLC) in the context of neoadjuvant immunotherapy. The authors highlighting the importance of immune checkpoint inhibitors (ICIs) in improving pCR rates and survival outcomes when added to neoadjuvant chemotherapy. The debate regarding evolving definitions of resectability, particularly in the context of T4 and N2 disease, is both relevant and critical for clinical practice, reflecting the dynamic nature of treatment strategies in this field.
The review addresses the controversy surrounding adjuvant therapy after resection and provides a nuanced perspective on patient selection for adjuvant therapies. Focusing on salvage and next-line treatments for neoadjuvant therapy or progression after R2 resection is particularly valuable as it addresses a gap in the literature regarding management strategies for these complex scenarios.
Predictive biomarkers and models are identified as areas in need of further research, highlighting the importance of personalized treatment approaches. Talk of ongoing work and future directions is encouraging; this suggests that the field is rapidly evolving with the potential for significant advances in the treatment of resectable Stage III NSCLC.
However, there are a few areas where the review could be improved:
1- A more detailed discussion on the management of adverse effects associated with ICIs in the neoadjuvant setting would be helpful. Given the importance of maintaining patients' suitability for subsequent surgery, strategies to minimize and manage these effects are crucial.
2- Please add the latest meta-analysis evaluation studies in this environment (https://doi.org/10.3390/cancers16010156)
3- Although survival and pathological response rates are critical, the impact of neoadjuvant immunotherapy on patients' quality of life can also be taken into account in the review. This includes the potential benefits of reducing preoperative tumor burden and adverse effects that may affect patients' well-being.
Comments on the Quality of English LanguageMinor editing of English language required
Author Response
Dear Reviewer
Thank you for reviewing the manuscript. Please find below the response to the comments.
Comment 1: This review provides a comprehensive and updated overview of the treatment of resectable Stage III non-small cell lung cancer (NSCLC) in the context of neoadjuvant immunotherapy. The authors highlighting the importance of immune checkpoint inhibitors (ICIs) in improving pCR rates and survival outcomes when added to neoadjuvant chemotherapy. The debate regarding evolving definitions of resectability, particularly in the context of T4 and N2 disease, is both relevant and critical for clinical practice, reflecting the dynamic nature of treatment strategies in this field.
The review addresses the controversy surrounding adjuvant therapy after resection and provides a nuanced perspective on patient selection for adjuvant therapies. Focusing on salvage and next-line treatments for neoadjuvant therapy or progression after R2 resection is particularly valuable as it addresses a gap in the literature regarding management strategies for these complex scenarios.
Predictive biomarkers and models are identified as areas in need of further research, highlighting the importance of personalized treatment approaches. Talk of ongoing work and future directions is encouraging; this suggests that the field is rapidly evolving with the potential for significant advances in the treatment of resectable Stage III NSCLC.
Response 1: Thank you for the positive feedback.
Comment 2: However, there are a few areas where the review could be improved. A more detailed discussion on the management of adverse effects associated with ICIs in the neoadjuvant setting would be helpful. Given the importance of maintaining patients' suitability for subsequent surgery, strategies to minimize and manage these effects are crucial.
Response 2: That is a very valid point. We have added a section ‘Toxicity and Quality of life’ to discuss this issue (page 13-14, line 488-510).
Comment 3: Please add the latest meta-analysis evaluation studies in this environment (https://doi.org/10.3390/cancers16010156)
Response 3: We have added and referenced the meta-analysis (page 2, line 63).
Comment 4: Although survival and pathological response rates are critical, the impact of neoadjuvant immunotherapy on patients' quality of life can also be taken into account in the review. This includes the potential benefits of reducing preoperative tumor burden and adverse effects that may affect patients' well-being.
Response 4: QoL is a very pertinent point. We have added a brief section (4.7) on toxicity and QoL with a reference of HRQoL data from CheckMate 816 (page 13-14, line 488-510).